# Connective Tissue Growth Factor in Idiopathic Pulmonary Fibrosis: Breaking the Bridge

**DOI:** 10.3390/ijms23116064

**Published:** 2022-05-28

**Authors:** Wiwin Is Effendi, Tatsuya Nagano

**Affiliations:** 1Department of Pulmonology and Respiratory Medicine, Faculty of Medicine, Universitas Airlangga, Surabaya 60132, Indonesia; 2Department of Pulmonology and Respiratory Medicine, Universitas Airlangga Teaching Hospital, Surabaya 60015, Indonesia; 3Pulmonology and Respiratory Medicine of UNAIR (PaRU) Research Center, Universitas Airlangga Teaching Hospital, Surabaya 60015, Indonesia; 4Division of Respiratory Medicine, Department of Internal Medicine, Kobe University Graduate School of Medicine, 7-5-1 Kusunoki-cho, Chuo-ku, Kobe 650-0017, Japan; tnagano@med.kobe-u.ac.jp

**Keywords:** CTGF, pro-fibrotic, mitochondria dysfunction, metabolic dysregulation, senescence, chronic respiratory diseases, idiopathic pulmonary fibrosis

## Abstract

CTGF is upregulated in patients with idiopathic pulmonary fibrosis (IPF), characterized by the deposition of a pathological extracellular matrix (ECM). Additionally, many omics studies confirmed that aberrant cellular senescence-associated mitochondria dysfunction and metabolic reprogramming had been identified in different IPF lung cells (alveolar epithelial cells, alveolar endothelial cells, fibroblasts, and macrophages). Here, we reviewed the role of the CTGF in IPF lung cells to mediate anomalous senescence-related metabolic mechanisms that support the fibrotic environment in IPF.

## 1. Introduction

IPF is a chronic, devastating, irreversible lung disease, characterized by injury-induced alveolar epithelial cell stress, progressive pathogenic myofibroblast differentiation, and imbalanced macrophage polarization, resulting in ECM deposition. Indeed, although other cell types undoubtedly contribute significantly, fibroblasts, the alveolar epithelium, and alveolar macrophages are the most critical drivers involved in the initiation and progression of pulmonary fibrosis. Cellular interaction between those intrinsically pathological cells, cell–matrix cross-talk, and abnormal immune activation, contributes to the multifarious pathogenesis of chronic progressive fibrotic diseases [1]. 

Normally, in wound repair, the myofibroblast secretes an ECM and undergoes apoptosis when the repair process is resolved. In IPF, the extensive accumulation of an immortal, myofibroblast-produced ECM results in lung stiffness that will maintain dynamic interplay between the ECM and other resident cells, such as fibroblasts, endothelial cells, pericytes, and fibrocytes [2]. The ECM affects structural remodeling and contributes to a fibrogenic niche in progressive fibrosis [3]. The cross-talk between the ECM and lung-resident cells plays a vital role in initiating fibrosis, which persists in disease progression [4].

Environmental exposure and occupational factors contribute to the risk of developing IPF. Even though there is no known cause for IPF, and its onset appears to occur spontaneously, several pieces of evidence emphasize that alveolar epithelial injury induced by environmental triggers will result in lung fibrosis [5,6]. 

The overexpression of a matricellular protein, connective tissue growth factor (CTGF), is a hallmark of fibrosis in multiple tissues in the mediation of the pro-fibrotic effects of tumor growth factor-β (TGF-β) [7]. CTGF is a primary mediator of TGF-β-induced pulmonary fibrosis. CTGF was found to be overexpressed in fibrotic lesions in major organs, including the lung, kidney, liver, skin, cardiovascular system, gastrointestinal system, eye, and gingiva [8]. This review highlights the role of CTGF in regulating cellular senescence, the metabolic reprogramming–mitochondria dysfunction mediator, and the pro-fibrotic environment in IPF.

## 2. Structure, Regulation, and Function of CTGF

CTGF, also known as the cellular communication network 2 (CCN2), is a TGF-β-target gene and a member of the CCN family of secreted proteins that regulate matricellular protein [9]. Matricellular proteins are expressed at higher levels during physiological and pathological processes, with distinct functions that bind to multiple receptors, other growth factors, and proteases, modulating their activity and mediating cross-talk between the ECM and cells [10].

CTGF (or CCN2) is a cysteine-rich, heparin-binding protein containing 349 amino acids, with an apparent molecular weight of 36–38 kDa. CCN family members have six members of multifunctional proteins, labeled CCN1 to CCN6. The CCN acronym is composed of the first three proteins members of the family: Cyr61 (cysteine-rich protein 61), CTGF, and NOV (nephroblastoma overexpressed gene) [11]. CCN proteins have a typical modular structure with four conserved domains, i.e., insulin-like growth factor (IGF)-binding proteins (IGFBPs) next to a von Willebrand factor type C repeat (VWC) (both are N-terminal fragments) and thrombospondin type I repeat (TSR) next to a C-terminal cystine-knot (CT) (forming a C-terminal fragment together) [12]. These domains each have specific binding partners, including an IGF protein for IGFBP, the TGF-β family for VWC, specific integrins (α4β1, α5β1, α6β1, and ανβ3) and sulfated glycoconjugates for TSP, and heparin-sulfate-containing proteoglycans (HSPGs), such as syndecan 4 and perlecan for CT [13,14].

CTGF expression is regulated at the transcriptional, post-transcriptional, and translational levels by various physiological and pathological factors [15]. Directly or through cross-talk with cell surface receptors, such as TGF-β and angiotensin, external stimuli initiate signaling pathways that recruit transcription factors to the nucleus, inhibiting or stimulating the expression of CTGF [16]. The critical transcription factors for the regulation of CTGF expression were found to be SMAD2, Yes-associated protein (YAP)/transcriptional coactivator with PDZ-binding motif (TAZ)/transcriptional enhancer factor TEF-1 (TEAD), ETS proto-oncogene 1 (ETS-1), PI3K-AKT, Fox0, and mitogen-activated protein kinase (MAPK)/Id-1 [17,18,19,20].

The biological function of CTGF is that it binds to specific receptors to initiate signal transduction, directly binding cytokines, regulating their availability and activity, mediating the matrix turnover by binding to ECM proteins, and regulating the activity of cytokines and growth factors through modulation cross-talk between signaling pathways [16]. CTGF is expressed in mesenchymal cell lineage and mediates physiological tissue regeneration and pathological fibrosis via ECM deposition, fibroblast proliferation, matrix production, angiogenesis, and granulation tissue formation [15,21]. Depending upon the microenvironment condition and cell type, CTGF is involved in several pathologic processes such as carcinogenesis and tumor development [22], diabetes [23], neuromuscular disorders [24,25], systemic sclerosis [26], ocular diseases [27], cardiac fibrosis [28,29], renal fibrosis [30], liver fibrosis [31], and lung fibrosis [32]. The regulation of CTGF is described in Figure 1.

## 3. CTGF Maintains the Pro-Fibrotic Environment in IPF

A recent hypothesis in the understanding of the pathogenesis of IPF stated that aberrant epithelial and epithelial–mesenchymal cross-talk responses to chronic alveolar epithelial injury might induce fibrosis independently of inflammatory events [33,34]. Alveolar epithelial injury provides an epithelium-associated pro-fibrotic environment. Recurrent injuries lead to epithelial apoptosis and drive the aberrant activation of epithelial cells to transdifferentiate into fibroblast epithelial–mesenchymal transition (EMT) [35,36,37]. There are phenotype changes characterized by downregulated epithelial markers, such as E-cadherin, whereas fibroblast-specific genes, such as *α*-smooth muscle actin (*α*-SMA), N-cadherin, fibroblast-specific protein 1 (FSP-1), and type I collagen, are upregulated [38]. Myofibroblasts can also modulate epithelial apoptosis, preserving a pro-fibrotic environment [39]. As a result, bidirectional EMT cross-talk assists the pro-fibrogenic positive feedback loop, resulting in fibrosis progression rather than wound resolution [40]. AECs become “vulnerable and sensitive to apoptosis,” but myofibroblasts become “apoptosis-resistant and immortal” [41].

Usual interstitial pneumonia (UIP) is a histopathologic and radiologic hallmark pattern for IPF. It is characterized by variations in temporospatial heterogeneity fibrosis, the accumulation of fibroblasts (fibroblast foci), and subpleural and paraseptal honeycombing [42]. Vanstapel et al. showed high expression of CTGF in fibrotic regions of restrictive allograft syndrome (RAS) lungs [43]. Furthermore, CTGF was found to be upregulated in cultured fibroblasts [44], injured epithelial cells [45], bronchoalveolar lavage and lung tissue [43], and plasma [46]. CTGF is upregulated in patients with IPF as well as in pro-fibrotic mediators and pro-fibrotic environments that contribute to fibrogenesis [47]. CTGF likely maintains aberrant responses of alveolar epithelial cells, fibroblasts, and alveolar macrophages in the development and progression of IPF (Figure 1). Many studies have reported that CTGF plays direct and indirect roles in accelerated aging, mitochondria dysfunction, and metabolic reprogramming. 

### 3.1. Activated Alveolar Epithelial Cells Initiate a Cycle of Fibrosis through CTGF

The precise mechanism of how CTGF-related activated epithelial cells induce fibrogenesis remains poorly defined. Following environmental injury, alveolar epithelial cells trigger their apoptosis and become active by secreting pro-fibrotic factors TGF-β to attract fibroblasts [48]. Type II alveolar epithelial cells (AECII) undergo EMT induced by EGFR–RAS–ERK signaling via zinc finger E-box-binding homeobox 1 (ZEB1)–tissue plasminogen activator (tPA), which augments fibroblast recruitment and activation [49]. AECII and activated fibroblasts secrete CTGF via autocrine and paracrine secretion, which contributes to the capacity of injured alveolar epithelial cells undergoing EMT to promote fibroblasts’ migration and proliferation [50,51]. The knockdown of the CTGF gene was shown to attenuate inflammatory responses induced by silica in bronchial epithelial cells [52].

Kasai et al. showed that CTGF might play a role in mediating the EMT process initiated by TGF-β1 [53]. Conversely, Shi et al. did not find evidence of the involvement of CTGF in the process of EMT induction via TGF-β1 [54]. However, a recent study proved that the effects of paracrine in secreted CTGF play an essential role in the EMT-like transition of epithelial cells into mesenchymal cells [55]. Therefore, the deletion of CTGF in mice lung epithelial cells attenuated the fibrotic response to bleomycin [51].

CTGF-induced EMT requires complex multiple signaling pathways to augment fibroblast migration and activation. Xu et al. demonstrated that CTGF contributes to fibroblast activation and matrix protein accumulation via phosphoinositide 3-kinase (PI3K) [45]. Integrin-linked kinase (ILK)-mediated CTGF was shown to induce EMT in AECII cells [56]. Cheng et al. reported that hypoxia-induced CTGF generated α-SMA and collagen expression via the MAPK–MAPK kinase (MEK)–extracellular-signal-regulated kinase (ERK) pathway [57]. Even though the role of ERK is unclear, the activation of the ERK signaling pathway in TGF-β1-induced EMT is crucial [58]. In addition, TGF-β-induced CTGF induces EMT-like changes in the adjacent epithelial cells through ERK, ADAM17, RSK1, and C–EBPβ pathways [59]. Therefore, the inhibition of the MAPK–MEK–ERK pathway might prevent the progression of pulmonary fibrosis [60]. The administration of CTGF was also followed by upregulated tenascin C, an element involved in modulating ECM integrity and cell physiology [61,62]. 

### 3.2. CTGF Stimulates the Differentiation of Lung Fibroblasts

Fibroblasts are tissue mesenchymal cells that are fundamental in establishing and maintaining an ECM. Fibroblast migration and activation, followed by myofibroblast differentiation, is the central pathogenesis of pulmonary fibrosis [63,64]. TGF-β regulates the mechanism of myofibroblast differentiation and connective tissue formation during physiological repairment and fibrotic processes. CTGF acts as a downstream mediator of TGF-β action, but CTGF does not act as a direct mediator to induce myofibroblast differentiation and collagen matrix contraction [65,66]. Several studies reported that CTGF triggered fibroblast proliferation and migration and myofibroblast differentiation [67,68,69]. The deletion of CTGF reduced ECM production, characterized by the low expression of COL1α2, COL3, and EDA-fibronectin mRNA [70]. 

As described previously, myofibroblasts may enhance the apoptosis of AECII. Although the primary source of oxidative stress is inflammatory cells, myofibroblasts generate reactive oxygen species (ROS) [71]. Shibata et al. demonstrated that secreted protein acidic and rich in cysteine (SPARC) promotes hydrogen peroxide (H_2_O_2_) secretion by TGF-β, leading to epithelial apoptosis [72]. Previously, Wang et al. demonstrated that the expression of CTGF and SPARC were increased in fibroblasts; therefore, SPARC might regulate the collagen expression by affecting the expression of CTGF [73]. Next, SPARC and CTGF seemed to be involved in the same biological pathway that upregulated collagen expression in mice fibroblasts [74]. 

### 3.3. CTGF Modulates Dysfunction of Macrophage Polarization 

Macrophage homeostasis is needed in the early phases of injury and the resolving phase. In IPF, there is an aberrant wound-healing process following an alveolar epithelial injury that involves the alteration of the polarization of M1 macrophages (pro-inflammatory) and M2 macrophages (anti-inflammatory) [75]. The continuous release of various pro-inflammatory cytokines and chemokines (M1 phenotypes) will preserve the fibrotic environment and induce the secretion of anti-inflammatory/pro-fibrotic cytokines (M2 phenotypes), leading to aberrant wound healing and tissue repair [76].

CTGF-associated macrophages drive polarization. CTGF was shown to be involved in the mechanism of an increase in M1 and a decrease in M2 macrophage markers in the pancreas [77]. Wang et al. also proved that CTGF regulates the polarization of macrophages in hepatocellular cells [78]. Furthermore, Zhang et al. revealed that the secretion of CTGF by M2 macrophages promotes fibroblast proliferation, migration, adhesion, and ECM production via activating the AKT–ERK1/2–STAT3 pathway in lung fibrosis [79]. Therefore, a CTGF blockade abolished M2-polarized macrophage influx [80].

### 3.4. CTGF Increases Endothelial Growth 

Although the mechanisms are not entirely clear, a study reported the possibility of endothelial cells being a source of myofibroblasts and undergoing endothelial–mesenchymal transition (EndoMT) [81]. The increased proliferation of endothelial cells was followed by fiber formation and ECM deposition via sterol regulatory element-binding protein 2 (SREBP2) [82]. Moreover, protein C3ar1 and galectin-3 induced EndoMT in vivo and in vitro [83].

CTGF regulated endothelial cell function and angiogenesis under certain pathological conditions [84,85]. CTGF interacted directly with vascular endothelial growth factor (VEGF) in driving the development of fibrosis and associated lymphangiogenesis/angiogenesis [86,87]. Kato et al. found that the level of CTGF protein was higher in bleomycin-treated mouse lungs than those in saline-treated lungs [88]. It was revealed that CTGF helps the transition of endothelial cells in EndoMT through direct and direct interaction with other pro-fibrotic proteins via hypoxia or inflammatory factors. 

### 3.5. Fibrocyte Differentiation Involved in CTGF 

Fibrocytes are the precursors of fibroblasts. The expression of fibrocytes in patients with IPF was high, but the expression of lung fibrocytes was significantly higher compared with circulating fibrocytes [89]. The association between the increased number of circulating fibrocytes and the mechanism of fibrocyte differentiation remains unclear. 

However, several studies support the involvement of CTGF in fibrocyte differentiation. CTGF contributes to fibrocyte proliferation and enhances fibrocyte differentiation into a myofibroblast phenotype through SMAD2 and ET_A_ receptor (ET_A_R) [90,91]. Under hypoxia conditions, CTGF was shown to induce the expression of circulating fibrocytes through hypoxia-inducible factor-1α (HIF-1α) and histone deacetylase 7 (HDAC7) [92].

## 4. CTGF Drives Senescence in IPF

Aging is one of the most critical risk factors, and many of the hallmarks of the aging lung (genomic instability, epigenetic alterations, telomere deterioration, loss of proteostasis, dysregulated nutrient sensing, mitochondrial dysfunction, cellular senescence, stem cell tiredness, modified intercellular communication, and ECM dysregulation) have been proposed as essential triggers for the development of IPF [93]. Recurrent microinjury in aging epithelial cells in genetically susceptible individuals leads to the aberrant activation of fibroblasts, resulting in the accumulation of ECM and fibrosis [94]. 

Both alveolar epithelial and fibroblast senescence trigger development and drive the progression of IPF. Lung fibroblast senescence was shown to reduce proliferation, increase migration, and induce cell-cycle arrest in alveolar epithelial cells [95]. Yao et al. recently demonstrated that AECII cells isolated from IPF lung tissue exhibit characteristic transcriptomic features of cellular senescence and promote progressive fibrosis [96]. In turn, specific molecular-signaling-pathway-associated senescent fibroblasts promote the occurrence and development of IPF [97]. Signal transducer and activator of transcription 3 (STAT3) activation pathways were involved in lung-fibrosis-driven fibroblast senescence [98]. Therefore, the clearance of cellular (fibroblast, epithelial, and endothelial) senescence improved lung function and fibrosis resolution [99]. 

Interestingly, the mitochondrial dysfunction of different cells shows different characteristics in pulmonary fibrosis. A schematic of how CTGF drives senescence-associated mitochondria dysfunction and metabolic dysregulation is depicted in Figure 1. 

### 4.1. CTGF Influences Cellular Mitochondria Bioenergetics

Mitochondria dysfunction and metabolic dysregulations are pathognomonic in IPF. The maintenance failure of mitochondrial quality control through three different mechanisms: (1) mitochondrial biogenesis; (2) mitochondrial dynamics (fusion and fission); and (3) mitophagy results in declined adenosine triphosphate (ATP) production, upregulated endoplasmic reticulum (ER) stress, and increased mitochondrial ROS (mtROS) [100]. Mitochondria dysfunction and metabolic reprogramming were identified in different IPF lung cells (alveolar epithelial cells, fibroblasts, and macrophages) to promote low resilience and increased susceptibility to the activation of pro-fibrotic responses [101]. Mitophagy or the selective degradation of mitochondria via autophagy, essential for the clearance of dysfunctional mitochondria, is downregulated in IPF [102]. 

Mitochondrial fusion proteins, such as mitofusin1 (MFN1) and mitofusin2 (MFN2), regulate surfactant lipids in AECII. However, the role of mitofusin is controversial. MFN2 was linked to the high production of ROS [103]. Meanwhile, MFN1- and MFN2-deficient AEC2 cells were more susceptible to mitochondrial damage, leading to lung fibrosis [104]. Additionally, the inhibition of ER-stress-modulated MFN2, through the repression of the PERK–ATF4 pathway, attenuated fibrosis, characterized by decreased protein expressions of CTGF, TGF-β, and α-SMA [105]. Moreover, ER stress inhibition attenuated the increases in α-SMA, CTGF, and TGF-β expressions and apoptotic markers [106].

#### 4.1.1. Alveolar Epithelial Mitochondria Dysfunction

AECII cells have the highest number of mitochondria for the production of ATP. Evidence in AECII emphysema indicated that high mitochondrial superoxide/mtROS production leads to reduced mitochondrial fusion, contributing to mitochondrial damage [107]. The elevation of mtROS-induced mitochondrial dynamic imbalance was shown to trigger impaired mitophagy alveolar cells, and the release of ER stress was shown to lead to epithelial apoptosis via PTEN-induced putative kinase 1 (PINK1) [108]. Furthermore, AECII releases senescence-associated secretory phenotype (SASP) protein with pro-fibrotic effects [96,109]. 

CTGF is one of the SASP factors. AECII cells augmented the expression of CTGF in bleomycin-induced lung fibrosis [110]. The expression of CTGF was significantly elevated on the mRNA level, suggesting epithelial senescence in kidney fibrosis [111]. Furthermore, CTGF was shown to decrease mitochondrial metabolism, resulting in ER-stress-associated pro-fibrotic effects in cardiovascular fibrosis [112].

#### 4.1.2. Lung Fibroblast Mitochondria Dysfunction

In contrast with AECII, the impairment of mitochondrial control in fibrotic lung fibroblasts leads to apoptosis resistance. Increased mtROS in fibroblasts promotes the damage of mtDNA through TGF-β1-mediated NADPH oxidase 4 (NOX4) signaling, resulting in reduced mitochondrial biogenesis via downregulated Nrf2 expression, leading to reduced mitophagy and the promotion of apoptosis resistance [113]. CTGF induces fibroblast senescence and associated anti-fibrotic phenotypes via p53 and p16^INK4a^ [114]. Yang et al. revealed the involvement of CTGF in TGF-β1-mediated NOX4 signaling [115]. 

The expression of CTGF is upregulated in fibroblast-related senescence. Kim et al. firstly demonstrated that CTGF is a novel biomarker protein of cellular senescence in fibroblasts [116]. There was an increased secretion of TGF-β1 and CTGF in response to a senescent fibroblast-derived ECM [117]. The overexpression of CTGF in fibroblasts induced the upregulation of p21 (CIp1/WAF1), Cyclin D1, and p16^Ink4A^, leading to autophagy and senescence [118].

#### 4.1.3. Macrophage Mitochondria Dysfunction

In addition to fibroblasts, IPF alveolar macrophages also drive apoptosis resistance. A study that used IPF specimens and mice models revealed that the activation of Akt1-mediated ROS in alveolar macrophages resulted in mitophagy and apoptosis resistance [119]. Augmented mitochondrial biogenesis was also generated via the upregulation of peroxisome proliferator-activated receptor-γ-coactivator 1-α (PGC-1α), Jumonji domain-containing 3 (Jmjd3), and mitochondrial transcription factor A (TFAM) [113]. In addition, the metabolic reprogramming of macrophage mitochondria regulates the switching of M1 macrophages to M2 macrophages in lung fibrosis [120]. Diminished mitochondrial quality control results in augmented mitochondrial dysfunction, increases ROS, which leads to decreased ATP production, promotes intrinsic apoptosis, and lung macrophages polarize to pro-fibrotic phenotypes [113]. The pro-fibrotic M2 lung alveolar macrophages were not dependent on fatty acid oxidation and synthesis or lipolysis but on glycolysis [121]. 

However, the precise role of CTGF in cellular mitochondria bioenergetics is not fully understood. Indeed, CTGF plays a supporting role as an essential downstream mediator of TGF-β1-induced mitophagy. The interplay between IPF mitochondria-produced ROS and CTGF induces glycolysis and mitophagy, leading to apoptosis resistance in macrophages and fibroblasts. By contrast, the accumulation of mtROS inhibits mitophagy to promote alveolar epithelial apoptosis. 

### 4.2. CTGF Regulates Cellular Metabolic Dysregulation 

Metabolic dysregulation is a hallmark of fibrosis and function as a critical contributor to the pathogenesis of the disease. The metabolism of glucose, lipid, glutamine, and other fuel substrates, affects proliferation, differentiation, apoptosis, autophagy, senescence, and inflammation [122]. The aberrant metabolism regulation in IPF involved the degradation of adenosine triphosphate (ATP), the impairment of glutathione (GSH)/upregulated glutamate levels, an imbalanced proline–ornithine ratio, increased glycolysis, an imbalanced arginine metabolism, decreased heme and biliverdin levels, a decreased tricarboxylic acid (TCA) cycle, and a downregulated sphingolipid metabolism [123,124].

Multiple cell types, including alveolar epithelial cells, fibroblasts, and macrophages, undergo the dysregulation of cellular metabolism. Environmental injury to AECII in a genetically predisposed host induces an aberrant cellular metabolism response within those cells, contributing to the development of pulmonary fibrosis. Fatty-acid-induced ER triggered AECII to become more vulnerable to apoptosis [125]. Dysregulated metabolic pathways of activated IPF fibroblasts result in upregulated glycolysis, lactate production, and increased glutamine metabolism [126]. In addition, macrophages generate ROS, use aerobic glycolysis to generate cytokines, and employ mitochondrial respiration to maintain inflammation, leading to fibrosis [127]. 

#### 4.2.1. Glucose Metabolism

Glucose metabolism deviation is essential for the development of fibrosis. Usually, the final process of glycolysis for the utilization of acetyl-CoA in mitochondria is followed by the conversion of pyruvate to lactic acid in low-oxygen-tension conditions. However, metabolic adaptation in cancer cells was shown to shift mitochondrial oxidative phosphorylation to aerobic glycolysis, which is known as the Warburg phenomenon [128]. Interestingly, the Warburg effect plays a crucial role in non-tumor diseases such as fibrosis. The metabolic shift illustrates the increased glucose uptake in the development of lung fibrosis. 

Aerobic glycolysis is an essential step in the initiation of the fibrotic process. A study demonstrated that reprogramming glucose metabolism, characterized by the elevation of the expression of glycolytic enzymes (6-phosphofructo-2-kinase/fructose-2, 6-biphosphatase 3 (PFKFB3), and HIF-1α) is required for the initiation and sustainment of myofibroblast differentiation [129]. The overexpression of CTGF in fibroblasts induces a “pseudo-hypoxic” state in mediating glycolysis via hypoxia-inducible factor-1α (HIF-1α) [118]. In turn, aerobic glycolysis was shown to sustain the YAP–TAZ signaling pathway, one of the most important transcription factors of CTGF [130]. Aerobic glycolysis and lactate production are prominent features of myofibroblast differentiation regulated by TGF-β, SMAD, and CTGF [131]. Hypoxia-induced CTGF expression contributes to pulmonary fibrosis via the mitogen-activated kinase–MEK kinase 1–extracellular-signal-regulated kinase 1–GLI 1–GLI2 and activator protein-1 (MEKK1–MEK1–ERK1–GLI 1–GLI 2 and AP-1) signaling pathways [57]. There is a mechanical connection between CTGF and enhanced aerobic glycolysis.

Furthermore, areas of fibroblastic foci in IPF are not just “passive” scar tissue but represent the “active” regions and the source of the upregulation of glycolytic pathways, resulting in the transformation of myofibroblast and ECM deposition [57]. Fibrotic foci in the honeycomb area are considered sites of ongoing lung injury with fibroproliferation, and fibroblasts and myofibroblasts are responsible for ECM deposition [132]. A high rate of aerobic glycolysis in fibrotic foci would influence myofibroblast differentiation and ECM production [133]. Emerging evidence revealed that the expression of CTGF was high and was involved in the appearance of the fibroblastic foci [134]. Moreover, glucose uptake in fibrotic foci was associated with glucose transporter-1 (GLUT-1) [133,134]. GLUT-1 mediates mesangial cell glucose flux, which leads to the activation of angiotensin II (Ang II), TGF-β, CTGF, and VEGF [135]. CTGF binds to integrin αvβ3, activating the FAK–Src–NF-κB p65 signaling axis, which results in the upregulation of GLUT3-mediated cell proliferation, migration, and glucose metabolism [136].

CTGF expression is an essential mediator of ECM protein expression in response to hyperglycemia [137]. Under a hyperglycemic environment, potent pro-fibrotic factors such as TGF-β and CTGF will modulate ECM production [138]. TGF-β1 and CTGF were also elevated in a mouse model of dermal fibrosis induced by the injection of bleomycin [139]. TGF-β signaling via CTGF enhances the glycolysis rate in human lung fibroblasts. TGF-β1 treatment increases glycolysis, glycolytic capacity, and oxygen consumption in WI-38 human lung fibroblasts [140]. 

Mesangial cells, which may acquire myofibroblast characteristics, were shown to increase the expression of TGF-β1 and CTGF when exposed to high levels of glucose [141]. Indeed, the expression of TGF-β1, SMAD3, SMAD7, and CTGF was upregulated significantly in high glucose concentrations of kidney fibrosis [142].

The overexpression of CTGF in IPF lungs is associated with the upregulation of aerobic glycolysis in cells in driving myofibroblast differentiation. Additionally, CTGF regulates glucose uptake in fibrotic foci as a fuel to maintain ECM accumulation and fibrotic lesions. The IPF lung resident cell complexity reflected metabolic reprogramming upon fibrosis. The fibroblast changes tend to benefit the upregulation of glycolytic pathways, while alveolar epithelial cells shift towards a shift in lipid metabolism [101]. Indeed, mitochondrial fusion and lipid metabolism are tightly linked to regulating cell-injury-associated AEC2 aberrant response [104].

#### 4.2.2. Lipid Metabolism

The majority of lipids stored in the human body are triglycerides, cholesterol, free fatty acids (FFA), and plasma membranes. The dysregulation of the lipid and FFA metabolism plays an essential role in the pathogenesis of IPF. However, its precise role in the pathology of IPF remains unclear. Alterations in lipid metabolism have been identified in IPF patients and animal models of lung fibrosis. Lipid metabolism is associated with glucose metabolism, as acetyl-CoA can be converted into lipids. A metabolomic study in IPF lung tissues showed an accumulation of FFA (palmitoleate, caproate, and myristate) and declined carnitine shuttle (palmitoylcarnitine, hexanoylcarnitine, and octanoylcarnitine) [124]. Additionally, a lipidomic study in IPF patients found the representation of eight types of lipid species (fatty acid, glycerolipid, saccharolipid, polyketide, sphingolipid, sterol lipids, prenol lipid, and glycerophospholipid) in IPF plasma [143]. 

Aberrations to the metabolism of lysophospholipids contribute to pulmonary fibrosis. Lysophosphatidic acid (LPA) and sphingosine 1-phosphate (S1P) are bioactive lysophospholipids that are involved in the differentiation of fibroblasts to myofibroblasts and EMT pathways [144]. Additionally, LPA and SP1 promote TGF-β activation, prevent apoptosis in fibroblasts, induce epithelial apoptosis, and increase vascular permeability [104]. 

The role of CTGF in lysophospholipid-driven fibrosis is enigmatic. CTGF was shown to be required for S1P-induced endothelial cell migration and angiogenesis [145]. Therefore, the inhibition of S1P induces the impaired expression of CTGF, leading to the amelioration of fibrosis [146]. The knockout of S1P3 receptor signaling significantly decreased CTGF levels and attenuated inflammation and fibrosis in a bleomycin-induced lung injury mice model [147]. There are various ways that the CTGF-mediated lysophospholipids signaling pathway is induced by several transcription activators. S1P agonists via receptors S1PR1 and S1PR3 were shown to cause a robust stimulation of ECM synthesis and expression of CTGF in normal human lung fibroblasts [148]. Cheng et al. showed that S1P promoted hepatocellular carcinoma (HCC) cell proliferation by upregulating CTGF expression through S1P2-mediated YAP activation [149]. TGF-β2-dependent upregulation requires S1P5 to induce pro-fibrotic CTGF [150]. S1P induced pro-fibrotic marker gene expression, including CTGF, via SMAD-independent pathways [151]. 

Furthermore, the newest study in this field demonstrated that the SphK1–S1P signaling axis is emerging as a critical player in developing IPF through the Hippo–YAP1 pathway [152]. The activation of LPA1 on mesothelial cells induced the expression of CTGF, driving fibroblast proliferation [153]. In response to LPA, CTGF was involved in the creation of ECM-produced fibroblasts via the integrin–focal adhesion kinase (FAK) signaling pathway [154]. Again, Yu et al. demonstrated that LPA augmented CTGF expression in osteoblasts via the activation of protein kinase C (PKC) and protein kinase A (PKA) [155]. The signaling molecules of CTGF in cross-talk and integration with transcription factors rely on the cell type and the physiological or pathological process involved.

The role of FFA is not well-described yet, although the levels of long-chain and medium-chain fatty acids are increased in IPF patients. Alterations in the FFA metabolism contribute to epithelial ER stress, apoptosis, EMT, secretion with pro-fibrotic signaling, and M2 polarization [156]. The elevation levels of FFAs may affect pulmonary fibrosis by regulating the TGF-β1-induced activation and proliferation of fibroblasts [157]. Through an in vivo study, Deng et al. demonstrated the role of CTGF in mediating YAP1 and suppressing FFA-induced apoptosis [158]. Previously, high-glucose and palmitate-elevated CTGF mRNA led to induced apoptosis and/or hypertrophy in cardiac myocytes via tyrosine kinase A (TrkA) [159]. It has been shown that CTGF could act as the downstream factor of several transcription factors.

Apart from lipid and fatty acids, lipid mediators are also involved in the pathogenesis of IPF by regulating the exhibition of pro- and anti-fibrotic effects in IPF. Lipid mediators are a class of bioactive lipids that are derived from phospholipids, sphingolipids, and polyunsaturated fatty acids and can be divided into pro-inflammatory/anti-fibrotic lipid mediators (such as prostaglandins and leukotrienes) and specialized pro-resolving/pro-fibrotic lipid mediators (SPMs) (including lipoxins, resolvins, maresins, and protectins) [160]. Regardless of the cell types involved, lipid mediators play roles in the activation of myofibroblasts, the deposition of ECM, and the remodeling of lung architecture and fibrosis [161]. 

However, the role of prostaglandin-driven fibrosis is a mystery. The expression of prostaglandin E2 (PGE2) in the BAL and lung tissue of IPF patients was low [162]. Additionally, PGE2 blocked TGFβ1-stimulated CTGF through c-Jun NH2-terminal kinase (JNK) [163] and Akt and Ca^2+^/calmodulin-dependent protein kinase-II (CaMK-II) [164]. By contrast, prostaglandin F2α (PGF2α) stimulated fibroblasts’ proliferation and collagen production independently of TGFβ1-stimulated CTGF [165].

#### 4.2.3. Glutamine Metabolism

The exact role of glutamine in myofibroblasts is unclear. However, emerging evidence shows aberrant glutamate, glutamine, and aspartate metabolisms in IPF. Enhanced glycolytic flux alone cannot fulfill the high metabolic demands of fibroblasts; hence, high levels of glutamine ensure sources of carbon and nitrogen to support cell growth [166]. The metabolic process by which glutamine is converted to glutamate by glutaminase (glutaminolysis) is required for TGF-β-induced collagen protein production in lung fibroblasts [167]. Glutaminolysis sustains the proliferation of the myofibroblast cell mass needed for energy metabolism and anabolism via YAP signaling [168]. The CTGF pathway was shown to be involved in glutamine regulation via glutamine synthetase (GS) [169].

In addition to cell proliferation, glutamine–glutamate metabolism regulates cell differentiation and apoptosis. The glutamine metabolism was shown to be involved in CTGF-induced cell differentiation in neural and retina cells [170,171]. Weiss et al. demonstrated that the glutamatergic synapse controlled apoptosis and the degeneration of different retinal cells induced by CTGF [172]. Furthermore, glutaminolysis promotes the apoptosis resistance of IPF fibroblasts through the epigenetic regulation of XIAP and survival [173].

### 4.3. CTGF Promotes Mitochondria–Metabolic-Dysfunction-Related Cellular Senescence 

It is already known that both cellular senescence and mitochondrial dysfunction have been defined as classical hallmarks of the aging process. However, their relationship in the development of IPF has not been made clear. Mounting evidence shows that mitochondrial dysfunction includes increasing ROS production, decreasing mitochondrial biogenesis, and impairing mitochondrial mitophagy, potentially impacting fibrotic processes. In addition, the cellular-senescence-associated apoptosis paradox contributes to the development of pulmonary fibrosis. Wiley et al. showed that mitochondrial dysfunction might induce cellular senescence via mitochondrial-dysfunction-associated senescence (MiDAS) [174]. Furthermore, mitochondria in senescent cells release damage-associated molecular patterns (DAMPs), which could promote the SASP [175]. 

CTGF is a SASP factor. We assume that CTGF may bridge mitochondrial-dysfunction-associated cellular senescence in IPF. Senescent IPF lung epithelial cells, fibroblasts, and myofibroblasts secrete CTGF as a pro-inflammatory SASP to induce senescence-associated fibrotic effects in surrounding cells. In epithelial cells, SASP is associated with gradual apoptotic cell loss in both the initiation and progression of fibrosis [96]. A recent study also showed that a CTGF-associated aberrant ECM contributes to fibrosis by inducing senescence [176]. Additionally, the overexpression of CTGF-induced cellular senescence in human airway epithelial cells was associated with the severity of airway obstruction among patients with smoking-induced COPD [177].

Moreover, the metabolic dysregulation of glucose, lipid, and glutamate is associated with senescence [178]. The overexpression of CTGF in IPF lungs is associated with the upregulation of aerobic glycolysis, aberrations to the metabolism of lysophospholipids, and glutaminolysis. Capparelli et al. demonstrated that the overexpression of CTGF was associated with the induction of glycolysis, mitophagy, and senescence phenotypes. A schematic of CTGF-regulated metabolic dysregulation and mitochondria dysfunction, which contribute to cellular senescence and the subsequent development of diseases, is depicted in Figure 2.

## 5. Conclusions and Future Perspectives

While research on remarkably different biological processes that might initiate inflammation and sustain pulmonary fibrosis is developing, much more work is required to hamper the progressivity of this disease. There is no definite cure for IPF. Two anti-fibrotic drugs that can delay the decline in lung function and improve quality of life have been approved to treat IPF—namely, nintedanib and pirfenidone. The pro-fibrotic properties of CTGF describe its ability to strengthen fibrogenic responses to other factors, such as TGF-β. Despite the complex signaling of CTGF that depends on the cell type and intricate nature of lung fibrosis, targeting CTGF regulation could be regarded as a targeted therapeutic for IPF. Strategies for hindering the fibrogenic actions of CTGF could be applied via pharmacological inhibitors, neutralizing antibodies, antisense oligonucleotides, or small interfering RNA (siRNA) [179]. 

The administration of FG-3019 (pamrevlumab), an anti-CTGF monoclonal antibody, was shown to suppress TGF-β1-induced fibroblast proliferation and myofibroblast differentiation and mesothelial to mesenchymal transition in IPF [180]. A phase 2, randomized, double-blind, placebo-controlled PRAISE trial for pamrevlumab, an anti-connective tissue growth factor therapy for idiopathic pulmonary fibrosis, showed optimal tolerability and comparable efficacy to current anti-fibrotic drugs and a reduction in the progression of fibrotic changes [181]. Discovering new pathways, including epigenetic mechanisms, such as the micro-RNA (mi-RNA) family, are essential in regulating CTGF and could be an attractive target treatment for IPF.

As this review highlights, CTGF drives the mechanism of aberrant response associated with an injury that may act directly or indirectly on matrix metabolism. The metabolic plasticity of resident cells allows metabolic reprogramming to maintain the high demands of the fibrotic process. Alveolar epithelial/fibroblast-derived CTGF acts as a mediator in linking signaling pathways of bidirectional EMT cross-talk and promotes fibroblasts’ proliferation. It has been demonstrated that CTGF regulates ROS production and alters mitophagy, which drives an imbalance of mitochondrial dynamics and apoptosis. The loss of mitochondrial control causes AECs to become sensitive to apoptosis, but myofibroblasts become apoptosis-resistant. Additionally, the high expression of CTGF is related to the dysregulation of macrophage polarization. 

Furthermore, specific signaling pathways in senescence-associated mitochondrial dysfunction involve CTGF stimulation that alters cellular metabolism. In cellular senescence, large amounts of CTGF production might provoke cellular metabolic-dysregulation-associated fibrosis. CTGF regulates glucose uptake in fibrotic foci to maintain ECM accumulation and fibrotic lesions, while alveolar epithelial cells, which are responsible for lipid metabolism and surfactant production, change lipid metabolism. In summary, CTGF is one of the best-studied pro-fibrotic factors that mediate other growth factors in the pathogenesis of pulmonary fibrosis in a downstream manner. CTGF precedes a signaling pathway from external and/or internal stimuli that mediate metabolic dysregulation and mitochondria dysfunction, which contributes to cellular senescence and leads to fibrotic processes.

## Figures and Tables

**Figure 1 ijms-23-06064-f001:**
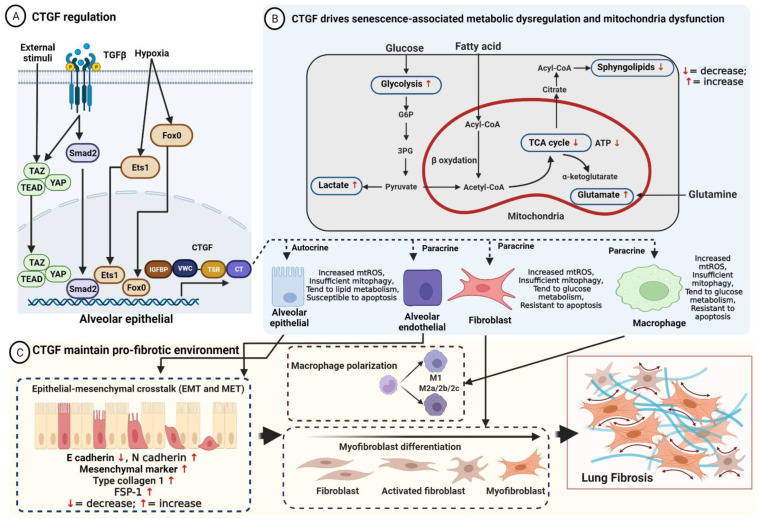
(**A**) Regulation of CTGF. CTGF expression is mainly regulated at the transcriptional level by various stimuli factors either directly or through cross-talk with cell surface receptors (TGF-β) that induce signaling pathways that recruit transcription factors (YAP/TAZ/TEAD, SMAD2, Ets-1, PI3K-AKT, and Fox0) to the nucleus, inhibiting or stimulating the expression of CTGF; (**B**) CTGF regulates aberrant metabolic responses associated with senescence of alveolar epithelial cells, endothelial cells, fibroblasts, and alveolar macrophages. As an essential downstream mediator of TGF-β1-induced mitophagy, CTGF induces mtROS and increases glycolysis, lactate, and glutaminolysis, leading to apoptosis resistance in macrophages and fibroblasts. Conversely, accumulation of mtROS inhibits mitophagy to promote alveolar epithelial apoptosis; (**C**) CTGF maintains pro-fibrotic environment. Injured AECII secretes CTGF via autocrine and paracrine, inducing alveolar epithelial cells undergoing EMT to promote fibroblasts’ migration and proliferation, regulating myofibroblast differentiation, and driving macrophage polarization, resulting in ECM deposition and lung fibrosis.

**Figure 2 ijms-23-06064-f002:**
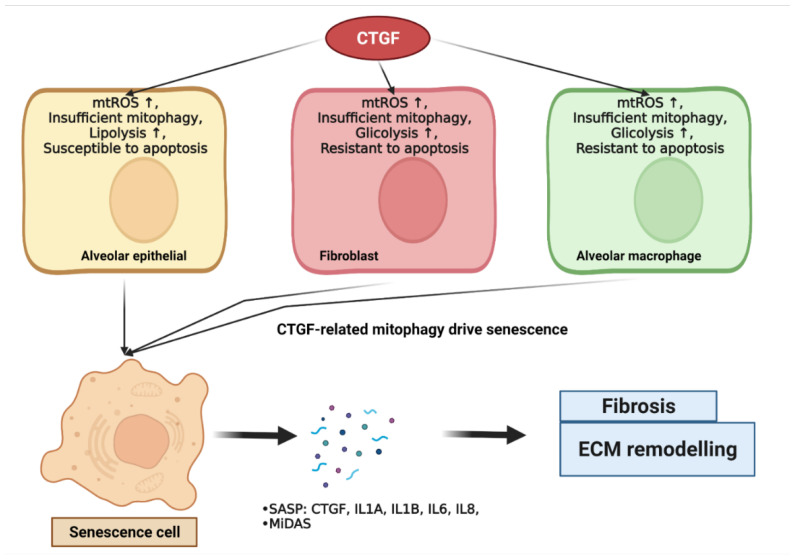
CTGF-regulated metabolic dysregulation and mitochondria dysfunction contribute to cellular senescence. CTGF drives metabolic- and mitochondria-dysfunction-associated mitophagy and contributes to cellular senescence. Briefly, CTGF expression in IPF cells leads to ROS production, metabolism disturbance, and paradoxical apoptosis, leading to mitophagy induction. Autophagy drives the onset of senescence. Senescent IPF lung epithelial cells, fibroblasts, and myofibroblasts secrete CTGF as pro-inflammatory SASP to induce senescence-associated fibrotic effects in surrounding cells via paracrine and autocrine signaling. Oxidative stress, autophagy, and senescence may also contribute to CTGF-induced fibrosis and ECM remodeling.

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
