# Peer review of "Connective Tissue Growth Factor in Idiopathic Pulmonary Fibrosis: Breaking the Bridge"

_ijms, 2022, doi:10.3390/ijms23116064_

Round 1
Reviewer 1 Report
This review is well-written and interesting. Particularly, they updated the role of CTGF in regulating aberrant metabolic responses and mitochondria dysfunction that are associated with senescence of alveolar epithelial, endothelial, fibroblasts, and alveolar macrophages. However, the weakness is that lack of significant discussion on how the CTGF-regulated metabolic responses and mitochondria dysfunction contribute to the cellular senescence and subsequent development of diseases. In addition, are there any preclinical and clinical trials concerned with CTGF as the therapeutic target?
Author Response
Review report (reviewer 1)
This review is well-written and interesting. Particularly, they updated the role of CTGF in regulating aberrant metabolic responses and mitochondria dysfunction that are associated with senescence of alveolar epithelial, endothelial, fibroblasts, and alveolar macrophages. However, the weakness is that lack of significant discussion on how the CTGF-regulated metabolic responses and mitochondria dysfunction contribute to the cellular senescence and subsequent development of diseases. In addition, are there any preclinical and clinical trials concerned with CTGF as the therapeutic target?
Author`s response
Thank You for the suggestion
- Lack of significant discussion on how the CTGF-regulated metabolic responses and mitochondria dysfunction contribute to the cellular senescence and subsequent development of diseases
We discussed the role of CTGF driving senescence that is associated with metabolic dysregulation and mitochondria dysfunction in sub-section 4.3. CTGF promotes mitochondria–metabolic-dysfunction-related cellular senescence
- Preclinical and clinical trials are concerned with CTGF as the therapeutic target.
We add clinical study and future therapy concerned with CTGF in IPF in lines 518-524.
Reviewer 2 Report
In this work Authors review the state of art of the biological/cellular mechanisms involved in idiopathic pulmonary fibrosis (IPF) and report an wide and exaustive literature about them. Unfortunately, it is not stressed the IPF as "Occupational disease" that is the topic of the special issue that this work should be inserted in. In the introduction section this aspect should be argued.
Authors have written in the recent past a plethora of reviews about the topic IPF, based on a consolidated framework, but unfortunately they did not and do not now contribute with their original experimental and/or clinical data, so this review (and also those written in the past) results a sterile work not adding new know-how in the field in the scientific community. Have the authors any evidence about the new pathways they envisage in the future perspective section? I think that introduction of novel results should be mandatory, since the biological mechanisms here reported are nowadays well known by insiders.
I further think that a graphycal abstract should be usefull to summarise the complexity of mechanisms.
Please, check spelling and abbreviations in the text and check english.
Author Response
Review report (reviewer 2)
In this work Authors review the state of art of the biological/cellular mechanisms involved in idiopathic pulmonary fibrosis (IPF) and report an wide and exaustive literature about them. Unfortunately, it is not stressed the IPF as "Occupational disease" that is the topic of the special issue that this work should be inserted in. In the introduction section this aspect should be argued.
Authors have written in the recent past a plethora of reviews about the topic IPF, based on a consolidated framework, but unfortunately they did not and do not now contribute with their original experimental and/or clinical data, so this review (and also those written in the past) results a sterile work not adding new know-how in the field in the scientific community. Have the authors any evidence about the new pathways they envisage in the future perspective section? I think that introduction of novel results should be mandatory, since the biological mechanisms here reported are nowadays well known by insiders.
I further think that a graphycal abstract should be usefull to summarise the complexity of mechanisms.
Please, check spelling and abbreviations in the text and check english.
Author`s response
- IPF as "Occupational disease.”
We highlighted occupational exposure as the factor that contribute to the risk of developing IPF (line 41-44).
- Publish article review and lack of an original article.
Authors confess this weakness and will improve in the future
- Any evidence about the new pathways they envisage in the future perspective section.
We discussed the role of CTGF to drive senescence that is associated with metabolic dysregulation and mitochondria dysfunction as a new pathway in sub-section 4.3. CTGF promotes mitochondria–metabolic-dysfunction-related cellular senescence.
Also, we re-emphasized these roles in lines 542-545 and figured it out in figure 2.
- Check spelling and abbreviations in the text and check English
We use the MDPI English editing service to revise the manuscript

Round 2
Reviewer 2 Report
The work is accepted in the revised form